# Wave Forces on a Partially Reflecting Wall by Oblique Bragg Scattering with Porous Breakwaters over Uneven Bottoms

Jen-Yi Chang [1] and Chia-Cheng Tsai [2,3,*]

[1] General Education Center, Tainan University of Technology, Tainan 710302, Taiwan; jimchang.taiwan@gmail.com
[2] Bachelor Degree Program in Ocean Engineering and Technology, National Taiwan Ocean University, Keelung 202301, Taiwan
[3] Center of Excellence for Ocean Engineering, National Taiwan Ocean University, Keelung 202301, Taiwan
[*] Correspondence: cctsai@mail.ntou.edu.tw

**Abstract:** In this study, the scattering of oblique water waves by multiple variable porous breakwaters near a partially reflecting wall over uneven bottoms are investigated using the eigenfunction matching method (EMM). In the solution procedure, the variable breakwaters and bottom profiles are sliced into shelves separated steps and the solutions on the shelves are composed of eigenfunctions with unknown coefficients representing the wave amplitudes. Using the conservations of mass and momentum as well as the condition for the partially reflecting sidewall, a system of linear equations is resulted that can be solved by a sparse-matrix solver. The proposed EMM is validated by comparing its results with those in the literature. Then, the EMM is applied for studying oblique Bragg scattering by periodic porous breakwaters near a partially reflecting wall over uneven bottoms. The constructive and destructive Bragg scattering are discussed. Numerical results suggest that the partially reflecting wall should be separated from the last breakwater by half wavelength of the periodic breakwaters to migrate the wave force on the vertical wall.

**Keywords:** partially reflecting wall; porous breakwater; step approximation; eigenfunction matching method; uneven bottom

## 1. Introduction

When considering the protection of harbors, wharfs, inlets, and shorelines from wave attacks, porous structures are frequently used as they can further dissipate wave energy. Therefore, the intensity of the wave energy on the shoreline decreases since only a small part of the wave energy is transmitted to the nearshore. Furthermore, coastal erosion and the corresponding coastal disasters are mitigated. Examples include seawalls, rubble-mound, subaerial, and submerged porous breakwaters. In this article, the combined effects of the partially reflecting vertical wall and multiple porous breakwaters on the coastal protection are studied.

Theoretical study of the energy dissipation inside porous structures was initialized by Sollitt and Cross [1], who evaluated the energy dissipation using the Lorentz's theory of equivalent work. Dalrymple, et al. [2] and Losada, et al. [3] applied this theory to compute the reflection and transmission coefficients of water wave scattering by subaerial porous breakwaters. In addition, Rojanakamthorn, et al. [4] and Rojanakamthorn, et al. [5] utilized the theory for water wave scattering by rectangular and trapezoidal porous submerged breakwaters, respectively. The theory of Sollitt and Cross [1] was widely applied for solving water wave scattering over porous breakwaters by the eigenfunction matching method [6–8] and the mild-slope equation [9–11]. These depth-integrated models are computationally efficient and can be served as preliminary calculations followed by modern three-dimensional numerical models [12–14] and/or experimental studies [15,16].

In practice, the shoreline is sometime protected by vertical breakwaters or seawalls which are usually considered as partially reflecting structures. Goda [17] studied several common vertical coastal structures and provided a list of the approximate partially reflecting factors, which vary from 0.3 to 1. In addition, Xiang and Istrati [18] found that the hydrodynamic forces depend on the ratio of the wavelength-to-width of the coastal structure. In order to model these vertical coastal structures, Isaacson and Qu [19] formulated a partially reflecting boundary condition. The condition was successfully applied for normal wave scattering by floating breakwaters in front of a harbor sidewall [20] and a submerged porous bar with a vertical wall [21]. In addition, Zhao et al. [22] and Behera and Khan [23] studied oblique wave scattering by a submerged porous bar near a partially reflecting wall and double trapezoidal porous breakwaters in front of a porous seawall, respectively. They found that multiple structures in presence of the vertical wall is the more efficient configurations in reducing the wave force on the wall. The later study was sequentially applied to other configurations with porous breakwaters [24–26].

Depth-integrated models are computationally efficient for solving problems of water wave scattering when comparing with the depth-resolved models based on the finite volume method [27], finite element method [28], and smooth particle hydrodynamics (SPH) [29–31]. These models include the mild-slope equation (MSE) [32], the eigenfunction marching method (EMM) [33], and the Miles' method [34,35]. The MSE has been successfully applied to solve problems of wave-seabed interactions [36,37], wave-current interactions [38], nonlinear waves [39], and wave-structure interactions [40]. A comprehensive review on the MSE can be found in a recent article by Porter [41]. Basically, the MSE is mainly applied for solving problems with variable bottoms. On the other hand, the EMM was initially applied as an analytical tool for obtaining solutions of problems with regular geometrical shapes [42–44]. By approximating the variable bottoms by shelves separated by steps, the EMM was used as a semi-analytical method for solving water wave scattering by variable structures over uneven bottoms in several previous studies [45–52]. The accuracy of the EMM solutions was demonstrated to be comparable to that of the MSE solutions [53]. In addition, the EMM is known to have a simpler mathematical formulation, as it requires no spatial derivatives of the eigenfunctions, which are needed in the MSE. However, the application of the EMM to three-dimensional, nonlinear, and/or time-dependent problems requires further investigation.

In the present study the EMM is formulated in the first time, to the best knowledge of the authors, for solving problems of water wave scattering by multiple variable porous breakwaters near a partially reflecting wall over uneven bottoms. By applying the conservation of mass and momentum to the eigen solutions on the shelves, the problems can be converted into a system of linear equations with unknown coefficients representing the wave amplitudes on the shelves. The sparse-matrix solver SuperLU was used to solve the resulting system [54]. The proposed EMM formulation is an extension of the traditional EMM for solving problems of water wave scattering by porous breakwaters resting on a flat bottom in front of totally reflecting [55,56] and partially reflecting [21,22] walls. Additionally, the proposed EMM model was validated by comparing with the solution by the boundary element method for water wave scattering by double trapezoidal porous breakwaters with a totally reflecting porous seawall [23]. Some discussions and further applications are also provided.

This paper is organized as follows. The wave problem is mathematically modeled and the EMM solution is developed in Section 2. The validations of the EMM model are provided in Section 3. Discussions and further applications are in Section 4. Finally, the conclusions of this study are presented in Section 5.

## 2. Materials and Methods

### 2.1. Problem Definition

In this subsection, the problem of water wave scattering by porous breakwaters near a partially reflecting vertical wall over uneven bottoms are formulated. Considering a train

of monochromatic water waves with incidence angle $\gamma$, amplitude $\bar{a}$, angular frequency $\sigma$, and wavelength $\lambda$, which propagate from left hand side toward porous breakwaters ended at the partially reflecting vertical wall over uneven bottoms. Here, the fluid is assumed to be irrational and incompressible and the surface waves are of small amplitude so that the linear wave theory can be applied [57]. A schematic configuration of the problem is depicted in Figure 1. In the figure, a Cartesian coordinate system is taken in which $z$-axis is chosen vertically upwards and the $x$-axis is taken as the rest position of the free surface. The wave motion is assumed to be time-harmonic by $e^{-i\sigma t}$, where $t$ is the time, $\mathbf{i}$ is the unit of complex numbers, and $\sigma$ is equal to $2\pi/T$ with $T$ being the wave period. In the step approximation, the porous breakwaters along with the uneven bottom are discretized into a series of $M-1$ shelves in the interval of $x_{m-1} \le x \le x_m$ for $m = 1, 2, \ldots, M-1$. The shelves are separated by $M-2$ steps at $x = x_m$ for $m = 1, 2, \ldots, M-2$. Additionally, $x_0 = -\infty$ is assumed and the partially reflecting vertical wall is located at $x = x_{M-1}$. On the $m$-th shelf, there is a porous layer located between $z = -d_m$ and $z = -h_m$, where $d_m$ and $h_m$ are the water and total depths, respectively. Therefore, the porous layer is absent if $d_m = h_m$ and it is subaerial if $d_m = 0$. Additionally, it is assumed that there are no porous media on the leftmost shelf, i.e., $d_1 = h_1$ in $x_0 \le x \le x_1$. Nothing is assumed on the rightmost shelf in $x_{M-2} \le x \le x_{M-1}$.

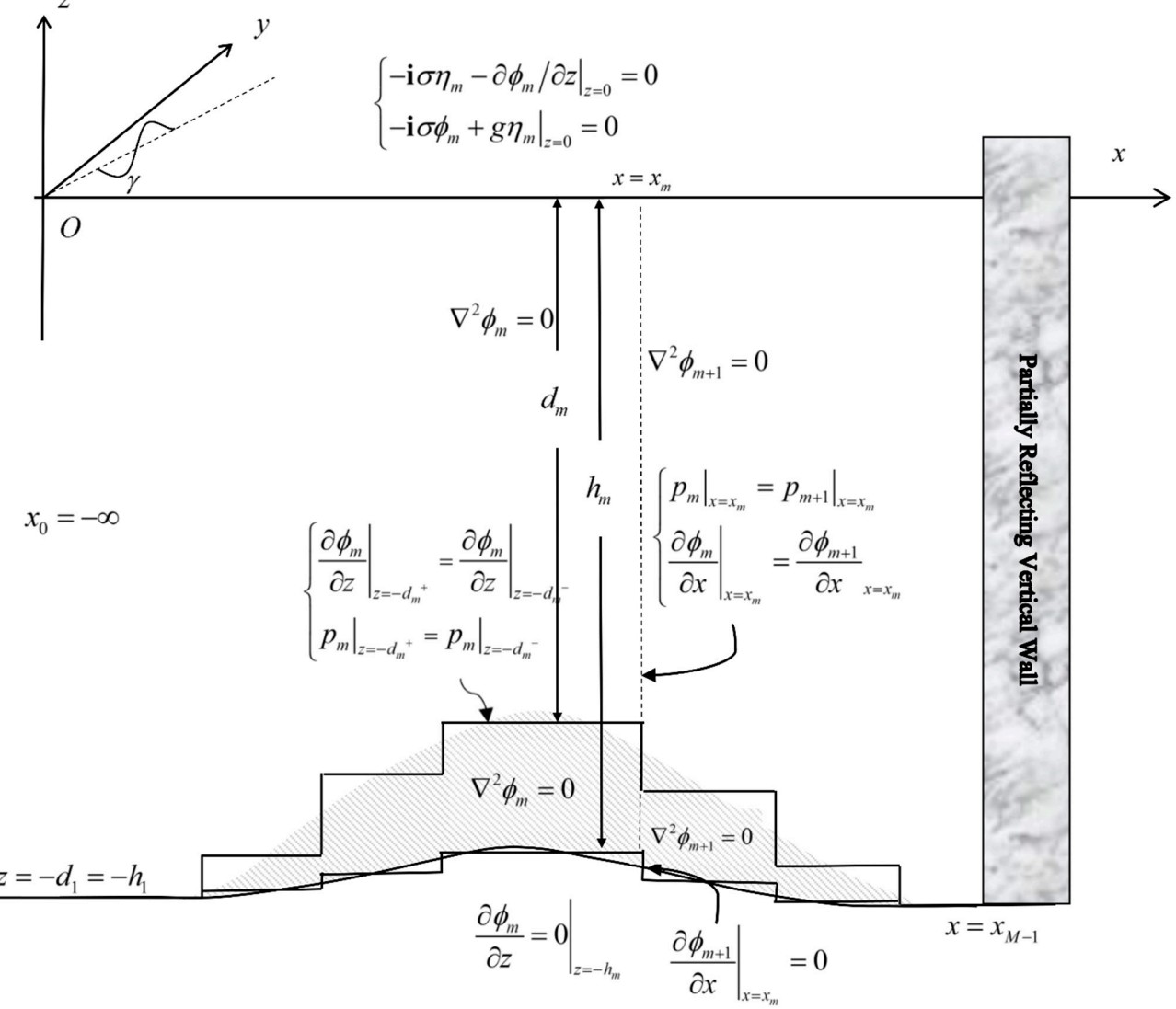

**Figure 1.** EMM definitions for the problem of water wave scattering by porous structures near a partially reflecting vertical wall over uneven bottoms.

Considering the solution on the $m$-th shelf in the interval $x_{m-1} \leq x \leq x_m$ for $m = 1, 2, \ldots, M - 1$, the velocity in the water layer $(-d_m \leq z \leq 0)$ and the discharge velocity in the porous layer $(-h_m \leq z \leq -d_m)$ are both defined by

$$\mathbf{u}_m = \nabla \phi_m, \tag{1}$$

where $\nabla = (\partial/\partial x, \partial/\partial y, \partial/\partial z)$ denotes the three-dimensional del operator with respect to the three-dimensional Cartesian coordinate $(x, y, z)$ and $\phi_m$ is the velocity potential. By the Bernoulli's equation [9], the pressures in water and porous layers are, respectively, defined as

$$p_m = -\rho(-\mathbf{i}\sigma\phi_m + gz) \tag{2}$$

and

$$p_m = -\rho\left(\frac{-\mathbf{i}\sigma s}{\varepsilon}\phi_m + gz + \frac{f}{\varepsilon}\sigma\phi_m\right) \tag{3}$$

with $\rho$, $g$, $\varepsilon$, $s$, and $f$ being the density of water, the acceleration of gravity, the porosity, inertial, and friction coefficients of the porous media, respectively. Following Dalrymple et al. [2], Losada et al. [58], and Twu et al. [6], $\varepsilon$, $s$, and $f$ were assumed to be known constants, whose effects were studied by parametric analyses. In this study, $s = 1$ is considered if not otherwise mentioned.

Sequentially by applying the continuity equation to Equation (1), the velocity potential is governed by the Laplace equations as

$$\nabla^2 \phi_m = 0, \tag{4}$$

which subjected to the kinematic and dynamic free-surface boundary conditions, respectively, as

$$-\mathbf{i}\sigma\eta_m - \frac{\partial\phi_m}{\partial z} = 0 \tag{5}$$

and

$$-\mathbf{i}\sigma\phi_m + g\eta_m = 0 \text{ on } z = 0, \tag{6}$$

where $\eta_m$ is the surface elevation. Equations (5) and (6) can be combined to obtain

$$\frac{\partial\phi_m}{\partial z} - \frac{\sigma^2}{g}\phi_m = 0 \tag{7}$$

In Equations (5)–(7) and the following of this section, it is assumed that the porous layer is submerged, i.e., $d_m > 0$, if it is not otherwise mentioned. Between the water and porous layers, the interface conditions are

$$\left.\frac{\partial\phi_m}{\partial z}\right|_{z=-d_m{}^+} = \left.\frac{\partial\phi_m}{\partial z}\right|_{z=-d_m{}^-} \tag{8}$$

and

$$p_m|_{z=-d_m{}^+} = p_m|_{z=-d_m{}^-}. \tag{9}$$

The bottom boundary condition is required to be

$$\frac{\partial\phi_m}{\partial z} = 0 \text{ on } z = -h_m. \tag{10}$$

Additionally, Equation (9) can be equivalently written as

$$\left.\frac{\phi_m}{\delta}\right|_{z=-d_m{}^+} = \phi_m|_{z=-d_m{}^-} \tag{11}$$

with $\delta$ being defined as

$$\delta = \frac{\varepsilon}{s + \mathbf{i}f}. \tag{12}$$

At the $m$-th step located at $x = x_m$ for $m = 1, 2, \ldots, M - 2$, the velocity potentials $\phi_m$ and $\phi_{m+1}$ require interface conditions

$$\left.\frac{\partial \phi_m}{\partial x}\right|_{x=x_m} = \left.\frac{\partial \phi_{m+1}}{\partial x}\right|_{x=x_m} \tag{13}$$

and

$$p_m|_{x=x_m} = p_{m+1}|_{x=x_m}, \text{ for } -\min(h_m, h_{m+1}) \le z \le 0, \tag{14}$$

where

$$\min(h_m, h_{m+1}) = \begin{cases} h_m \text{ if } h_m \le h_{m+1} \\ h_{m+1} \text{ if } h_m > h_{m+1}. \end{cases} \tag{15}$$

Equation (14) can be equivalently rewritten as

$$\Delta(\phi_m) = \Delta(\phi_{m+1}), \tag{16}$$

where

$$\Delta(\phi_m) = \begin{cases} \phi_m \text{ if } 0 \ge z \ge -d_m \\ \frac{\phi_m}{\delta} \text{ if } -d_m > z \ge -h_m. \end{cases} \tag{17}$$

Additionally, the condition for the vertical wall is described by

$$\frac{\partial \phi}{\partial x} = 0 \text{ for } -\max(h_m, h_{m+1}) \le z \le -\min(h_m, h_{m+1}), \tag{18}$$

where

$$\max(h_m, h_{m+1}) = \begin{cases} h_{m+1} \text{ if } h_m \le h_{m+1} \\ h_m \text{ if } h_m > h_{m+1} \end{cases} \tag{19}$$

and $\phi$ stands for either $\phi_m$ or $\phi_{m+1}$ depending on which side of the vertical wall is filled with water or porous media. Additionally, the partially reflecting condition of the vertical wall can be expressed as

$$\left.\left(\frac{\partial \phi_{M-1}}{\partial x} - \mathbf{i}\hat{k}_{M-1,0}\frac{1 - K_w}{1 + K_w}\phi_{M-1}\right)\right|_{x=x_{M-1}} = 0, \tag{20}$$

where $\hat{k}_{M-1,0}$ is the wavenumber to be defined in the next subsection and $K_w$ is *a priori* given as the partially reflecting factor of the vertical wall [19,21,22,57]. It needs to be mentioned that Equation (20) was originally developed without the effects of evanescent modes.

In order to make the solution unique, the far-field condition for the surface elevation of the incident wave is required as

$$\eta_1 = \bar{a}\left(e^{\mathbf{i}\hat{k}_{1,0}x} + K_R e^{\mathbf{i}\theta_R}e^{-\mathbf{i}\hat{k}_{1,0}x}\right)e^{\mathbf{i}k_y y} \text{ as } x \to -\infty, \tag{21}$$

where the reflection coefficient $K_R$ and the phase angle $\theta_R$ are real numbers so that $K_R = |K_R e^{\mathbf{i}\theta_R}|$. In Equation (21), $k_y$ and $\hat{k}_{1,0}$ are positive real wavenumbers to be defined in the next subsection.

### 2.2. Dispersion Relations and Eigenfunctions

By separating variables along with the eigenfunction expansion method, the complete solution of the velocity potential on the $m$-th shelf can be defined as

$$\phi_m(x, y, z) = \sum_{n=0}^{N}\left(A_{m,n}\xi_{m,n}^{(1)}(x) + B_{m,n}\xi_{m,n}^{(2)}(x)\right)\zeta_{m,n}(z)e^{\mathbf{i}k_y y} \tag{22}$$

for $m = 1, 2, 3, \ldots, M - 1$. Additionally $A_{m,n}$ and $B_{m,n}$ are unknown coefficients to be solved by the method addressed in the next subsection. To construct complete solutions by the method of the separation of variables, the eigenfunctions in Equation (22) can be expressed as

$$\zeta_{m,n}(z) = \begin{cases} \alpha_{1,m,n} e^{k_{m,n}z} + \alpha_{2,m,n} e^{-k_{m,n}z} & \text{for } z \leq -d_m \\ \beta_{1,m,n} e^{k_{m,n}z} + \beta_{2,m,n} e^{-k_{m,n}z} & \text{for } z \geq -d_m, \end{cases} \tag{23}$$

$$\xi_{m,n}^{(1)}(x) = e^{\mathrm{i}\hat{k}_{m,n}(x-\bar{x}_{m-1})}, \tag{24}$$

and

$$\xi_{m,n}^{(2)}(x) = e^{-\mathrm{i}\hat{k}_{m,n}(x-\bar{x}_m)} \tag{25}$$

with

$$\hat{k}_{m,n} = \sqrt{k_{m,n}^2 - k_y^2} \tag{26}$$

and

$$\begin{cases} \bar{x}_m = x_m & \text{for } m = 1, 2, \ldots, M - 1 \\ \bar{x}_0 = 0. \end{cases} \tag{27}$$

In Equation (22), $k_y$ is the transverse wavenumber of the incident wave as

$$k_y = k_{1,0} \sin \gamma, \tag{28}$$

where $k_{1,0} = 2\pi/\lambda > 0$ is the wavenumber of the incident wave. According to the Snell's law [59] and the linear wave theory [57], the transverse wavenumber $k_y$ is constant for $m = 1, 2, 3, \ldots, M - 1$. In Equations (24) and (25), $\hat{k}_{m,n}$ is the lateral wavenumber corresponding to the absolute wavenumber $k_{m,n}$ via Equation (26). Here, it needs to be mentioned that the wave directionality follows the law of reflection, i.e., the angle of reflection equals to the angle of incidence, from Equation (22) with $m = 1$ and $N = 0$. If the EMM with evanscent modes is considered to refine the prescribed result on wave directionality, the resulted surface elevations should be analyzed for obtaining the angle of reflection by physical or other computational methods [60–62]. This is ignored as the length of the article is already lengthy.

It can be noticed that the solutions defined by Equation (22) satisfy the governing Equation (4) analytically. Sequentially, the absolute wavenumbers $k_{m,n}$ can be determined by enforcing Equations (7), (8), (10), and (11) such that the unknown coefficients $\alpha_{1,m,n}$, $\alpha_{2,m,n}$, $\beta_{1,m,n}$, and, $\beta_{2,m,n}$ have nontrivial solutions. This gives the dispersion relation in the matrix form as

$$\det \begin{pmatrix} k_{m,n} - \frac{\sigma^2}{g} & -k_{m,n} - \frac{\sigma^2}{g} & 0 & 0 \\ e^{-k_{m,n}d_m} & -e^{k_{m,n}d_m} & -e^{-k_{m,n}d_m} & e^{k_{m,n}d_m} \\ e^{-k_{m,n}d_m} & e^{k_{m,n}d_m} & -\frac{e^{-k_{m,n}d_m}}{\delta} & -\frac{e^{k_{m,n}d_m}}{\delta} \\ 0 & 0 & e^{-k_{m,n}h_m} & -e^{k_{m,n}h_m} \end{pmatrix} = 0. \tag{29}$$

It can be proved that the prescribed dispersion relation is equivalent to the explicit form of the dispersion relation addressed in the literature [4,9].

If the wavenumbers $k_{m,n}$ can be solved from Equation (29), the unknown coefficients $\alpha_{1,m,n}$, $\alpha_{2,m,n}$, $\beta_{1,m,n}$, and, $\beta_{2,m,n}$ for the depth eigenfunciton (23) can be solved by using Equations (8), (11), (10), as well as the uniqueness condition

$$\zeta_{m,n}(z = 0) = 1. \tag{30}$$

This can be expressed in the matrix form as

$$
\begin{pmatrix}
1 & 1 & 0 & 0 \\
e^{-k_{m,n}d_m} & -e^{k_{m,n}d_m} & -e^{-k_{m,n}d_m} & e^{k_{m,n}d_m} \\
e^{-k_{m,n}d_m} & e^{k_{m,n}d_m} & -\dfrac{e^{-k_{m,n}d_m}}{\delta} & -\dfrac{e^{k_{m,n}d_m}}{\delta} \\
0 & 0 & e^{-k_{m,n}h_m} & -e^{k_{m,n}h_m}
\end{pmatrix}
\begin{pmatrix}
\alpha_{1,m,n} \\
\alpha_{2,m,n} \\
\beta_{1,m,n} \\
\beta_{2,m,n}
\end{pmatrix}
=
\begin{pmatrix}
1 \\
0 \\
0 \\
0
\end{pmatrix},
\tag{31}
$$

which can be numerically solved by the Gaussian elimination method [63].

When solving the porous dispersion relation (29), the Newton-Raphson method [5] is applied with the wavenumbers of water wave being adopted as the initial guesses. In addition, the perturbation technique is used to ensure the convergence [11,58,64]. Here, the wavenumbers of water waves without porous breakwaters ($d_m = h_m$) can be obtained by the dispersion relation as

$$
\frac{\sigma^2}{g} = k_{m,n}\tanh k_{m,n}h_m,
\tag{32}
$$

which admits a propagating wavenumber $k_{m,0}$ and a sequence of evanscent wavenumbers $k_{m,n}$ for $n = 1, 2, \ldots$. Correspondingly, the depth eigenfunciton (23) become

$$
\zeta_{m,n}(z) = \frac{\cosh k_{m,n}(z + h_m)}{\cosh k_{m,n}h_m}.
\tag{33}
$$

Details on the wavenumbers and depth eigenfunciton of water waves without porous breakwaters can be found in the literature [57]. These complete the introductions on the dispersion relations and eigenfunctions.

### 2.3. Subaerial Porous Breakwaters

When the porous layer is subaerial (i.e., $d_m = 0$), the free surface conditions (5)–(7) should be replaced by

$$
- i\sigma\varepsilon\eta_m - \frac{\partial\phi_m}{\partial z} = 0,
\tag{34}
$$

$$
-i\sigma\frac{(s + if)}{\varepsilon}\phi_m + g\eta_m = 0, \text{ on } z = 0,
\tag{35}
$$

and

$$
\frac{\partial\phi_m}{\partial z} - \frac{\sigma^2(s + if)}{g}\phi_m = 0.
\tag{36}
$$

Consequentially, the dispersion relation of waves in subaerial porous breakwaters becomes

$$
\frac{\sigma^2(s + if)}{g} = k_{m,n}\tanh k_{m,n}h_m.
\tag{37}
$$

The depth eigenfunciton can be derived to be the same as Equation (33). Details can be found in the literatures [2,3].

### 2.4. Eigenfunction Matching Method

The solution expression given in Equation (22) with Equations (23)–(29), and (31) satisfies analytically the governing Equation (4) as well as the vertical conditions (7), (8), (10), (11), and (30). The unknown coefficients $A_{m,n}$ and $B_{m,n}$ can be solved by the matching conditions (13), (16), and (18), the far-field condition (21) as well as the partially reflecting condition (20).

To be clearer, the conservation of mass in Equations (13) and (18) can be formulated as

$$
\left\langle \frac{\partial\phi_m}{\partial x} \middle| \zeta_{m,l}^{\text{large}} \right\rangle = \left\langle \frac{\partial\phi_{m+1}}{\partial x} \middle| \zeta_{m,l}^{\text{large}} \right\rangle \text{ for } m = 1, 2, \ldots, M - 2 \text{ and } l = 0, 1, \ldots, N,
\tag{38}
$$

where the inner product of two vertical eigenfunctions is defined by

$$\langle G_1 | G_2 \rangle = \int_{-\lambda}^{0} G_1 \Delta(G_2) dz, \tag{39}$$

where $G_1$ and $G_2$ are the vertical eigenfunction of $\zeta_{m,n}$ with arbitrary $m$ and $n$, as well as $\lambda$ denotes the total depth of the vertical eigenfunction $G_1$. Additionally, the functional $\Delta(G_2)$ is defined by Equation (17). The definition of the inner product ensures the orthogonal relation as

$$\langle \zeta_{m,n} | \zeta_{m,l} \rangle = 0 \text{ for } n \neq l. \tag{40}$$

In Equation (38), the vertical eigenfunction $\zeta_{m,l}^{\text{larger}}$ is defined by

$$\zeta_{m,l}^{\text{larger}} = \begin{cases} \zeta_{m,l} & \text{for } h_m > h_{m+1} \\ \zeta_{m+1,l} & \text{for } h_{m+1} > h_m. \end{cases} \tag{41}$$

The conservation of momentum in Equation (16) can be expressed as

$$\left\langle \zeta_{m,l}^{\text{small}} | \phi_m \right\rangle = \left\langle \zeta_{m,l}^{\text{small}} | \phi_{m+1} \right\rangle \text{ for } m = 1, 2, \ldots, M-2 \text{ and } l = 0, 1, \ldots, N, \tag{42}$$

where

$$\zeta_{m,l}^{\text{smaller}} = \begin{cases} \zeta_{m,l} & \text{for } h_m < h_{m+1} \\ \zeta_{m+1,l} & \text{for } h_{m+1} < h_m. \end{cases} \tag{43}$$

Here, it should be noted that Equations (38) and (42) are valid for all six cases even if the porous layer is subaerial or absent, as shown in Figure 2.

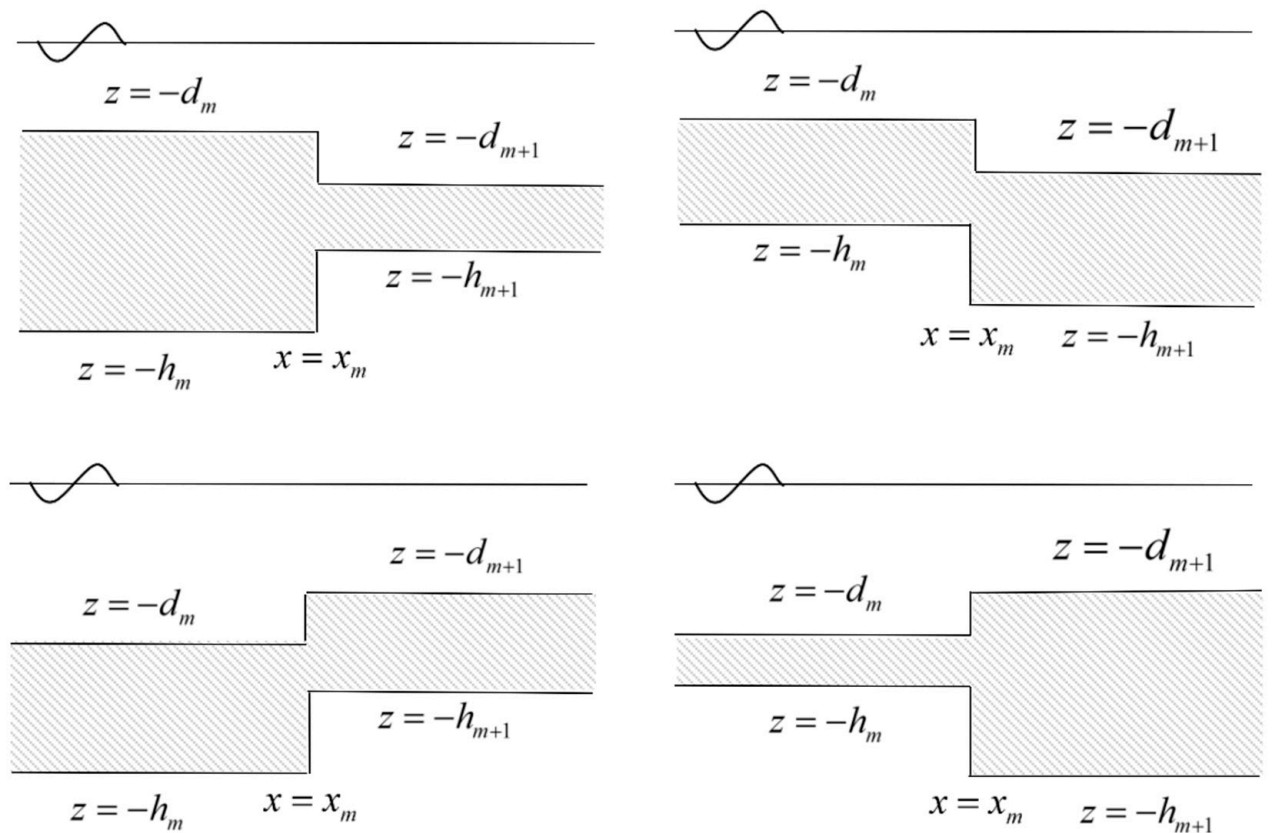

**Figure 2.** *Cont.*

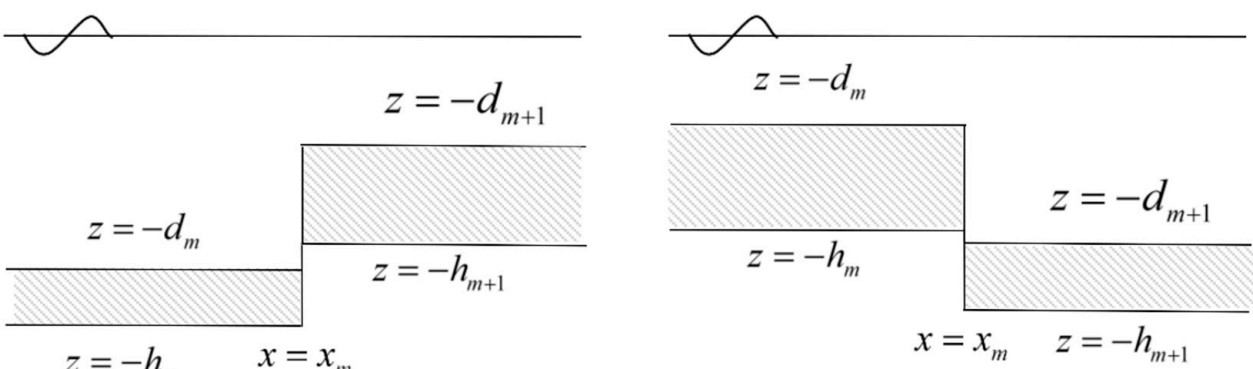

**Figure 2.** Schematics for six different situations of shelves separated by abrupt connections.

Then based on the far-field condition (21), the dynamic free-surface boundary condition (6), and Equation (30), the far-field solution of the velocity potential can be expressed as

$$\phi_1 = -\frac{i\bar{a}g}{\sigma}\zeta_{1,0}(z)\left(e^{i\hat{k}_{1,0}x} + K_R e^{i\theta_R}e^{-i\hat{k}_{1,0}x}\right)e^{ik_y y} \text{ as } x \to -\infty. \tag{44}$$

Substituting Equation (44) into Equation (22), we can obtain following equations

$$B_{1,0}e^{i\hat{k}_{m,n}\bar{x}} = -\frac{i\bar{a}K_R e^{i\theta_R}g}{\sigma}, \tag{45}$$

$$A_{1,0} = -\frac{i\bar{a}g}{\sigma}, \tag{46}$$

and

$$A_{1,n} = 0 \text{ for } n = 1, 2, \ldots, N. \tag{47}$$

In addition, the combination of the solution expression (22) and partially reflecting condition (20) gives

$$\sum_{n=0}^{N}\left(i\hat{k}_{M-1,n}A_{M-1,n}\xi_{M-1,n}^{(1)}(x_{M-1}) - i\hat{k}_{M-1,n}B_{M-1,n}\xi_{M-1,n}^{(2)}(x_{M-1})\right)\langle\zeta_{M-1,n}|\zeta_{M-1,l}\rangle$$
$$= i\hat{k}_{M-1,0}\left(\frac{1-K_w}{1+K_w}\right)\sum_{n=0}^{N}\left(A_{M-1,n}\xi_{M-1,n}^{(1)}(x_{M-1}) + B_{M-1,n}\xi_{M-1,n}^{(2)}(x_{M-1})\right)\langle\zeta_{M-1,l}|\zeta_{M-1,n}\rangle \tag{48}$$

for $l = 0, 1, \ldots, N$. Finally, the substitutions of solution expression (22) into the conservation of mas (38) and momentum (42), respectively, result in

$$\sum_{n=0}^{N}\left(i\hat{k}_{m,n}A_{m,n}\xi_{m,n}^{(1)}(x_m) - i\hat{k}_{m,n}B_{m,n}\xi_{m,n}^{(2)}(x_m)\right)\langle\zeta_{m,n}|\zeta_{m,l}^{\text{larger}}\rangle$$
$$= \sum_{n=0}^{N}\left(i\hat{k}_{m+1,n}A_{m+1,n}\xi_{m+1,n}^{(1)}(x_m) - i\hat{k}_{m+1,n}B_{m+1,n}\xi_{m,n}^{(2)}(x_m)\right)\langle\zeta_{m+1,n}|\zeta_{m,l}^{\text{larger}}\rangle \tag{49}$$

and

$$\sum_{n=0}^{N}\left(A_{m,n}\xi_{m,n}^{(1)}(x_m) + B_{m,n}\xi_{m,n}^{(2)}(x_m)\right)\langle\zeta_{m,l}^{\text{smaller}}|\zeta_{m,n}\rangle$$
$$= \sum_{n=0}^{N}\left(A_{m+1,n}\xi_{m,n}^{(1)}(x_m) + B_{m+1,n}\xi_{m,n}^{(2)}(x_m)\right)\langle\zeta_{m,l}^{\text{smaller}}|\zeta_{m+1,n}\rangle, \tag{50}$$

for $l = 0, 1, \ldots, N$ and $m = 1, 2, \ldots, M-2$. In summary, the EMM solution procedure can begin with Equations (46)–(50), which compose $2(M-1)(N+1)$ linear equations for solving the $2(M-1)(N+1)$ unknowns $A_{m,n}$ and $B_{m,n}$ if the matrix system of the linear equations is nonsingular. Then, Equation (45) can be used to solve the reflection coefficient

$K_R$. In this study, the SuperLU library is used for solving the sparse matrix of the resultant system of linear equations [54].

### 2.5. Wave Force on the Partially Reflecting Wall

The dynamic pressure on the vertical wall can be calculated by the Bernoulli's Equations (2) and (3) as

$$p_{M-1}|_{x=x_{M-1}} = \mathbf{i}\rho\sigma\Delta(\phi_{M-1})|_{x=x_{M-1}}. \tag{51}$$

Integrating the dynamic pressure along the vertical wall, we get the magnitude of dimensionless horizontal wave force with the normalization factor $2\bar{a}\rho g h_{M-1}$ as

$$K_F = \frac{\mathbf{i}\sigma \int_{-h_{M-1}}^{0} \Delta(\phi_{M-1})dz\Big|_{x=x_{M-1}}}{2\bar{a}g h_{M-1}}. \tag{52}$$

Sequential substituting the solution expression Equation (22) into the above equation gives the required formula for the dimensionless wave force on the vertical wall as

$$K_F = \frac{\mathbf{i}\sigma \sum_{n=0}^{N} \left( A_{M-1,n}\xi_{M-1,n}^{(1)}(x_{M-1}) + B_{M-1,n}\xi_{M-1,n}^{(2)}(x_{M-1}) \right) \int_{-h_{M-1}}^{0} \Delta(\zeta_{M-1,n}(z))dz}{2\bar{a}g h_{M-1}}. \tag{53}$$

In deriving Equation (53), the transverse wave function $e^{\mathbf{i}k_y y}$ is neglected as the equation is normalized by considering its maximum value in the transverse direction.

Here, the formulation is derived for the in-plane horizontal wave force, other hydrodynamic parameters, such as the out-of-plane horizontal force, the overturning, and yaw moments can be derived by following the previous study [62]. These will be the topics of future research as the article is already lengthy.

## 3. Results

In this section, the EMM model is validated by comparing its results with those in the literature. Several cases of water wave scattering by porous structures near a partially reflecting wall are considered. The convergence is carefully studied by examining the test cases.

### 3.1. A Rectangular Porous Structure near a Partially Reflecting Vertical Wall

Following Zhao et al. [21], let's consider the problem of water wave scattering by a rectangular porous breakwater near a vertical wall over a uniform bottom with $\gamma = 0$, $K_w = 0.5$, $D/h_1 = 1$, $b/h_1 = 1$, $d_2/h_1 = 0.5$, $\varepsilon = 0.45$, and $f = 2$ as depicted in Figure 3. To study the convergence with respect to the number of evanescent modes $N$, the reflection coefficient $K_R$ is plotted against the dimensionless wavenumber $k_{1,0}h_1$ as depicted in Figure 4. In the figures, it can be noted that the convergence is achieved for evanescent modes increase up to $N = 10$. Additionally, the results are in good agreements with those obtained by Zhao et al. [21]. Numerical reflection coefficients are detailed in Table 1.

The study is then extended to consider a problem with oblique incidence. Here, we consider $\gamma = 30^o$, $K_w = 0.5$, $D/h_1 = 3$, $b/h_1 = 0.8$, $d_2/h_1 = 0.2$, $\varepsilon = 0.4$, and $f = 2$. Figure 5 shows the reflection coefficient $K_R$ varying against the dimensionless wavenumber $k_{1,0}h_1$ for different numbers of evanescent modes $N$. Convergences can also be observed to be significant. In addition, the results also agree well with those in the literature [22].

Overall, these two numerical examples demonstrate that the EMM can be applied to solve problems of water wave scattering by a rectangular porous breakwater near a partially reflecting vertical wall with normal or oblique incidences.

### 3.2. Wave Force on the Vertical Wall

An important marine problem is to determine of the forces exerted on a vertical wall when waves are reflected by the wall. Zhao et al. [21] studied the wave force for normal incident wave scattering by a porous rectangular breakwater near a partially reflecting vertical with $\gamma = 0$, $K_w = 0.5$, $D/h_1 = 3$, $b/h_1 = 1.2$, $d_2/h_1 = 0.2$, $\varepsilon = 0.4$, and $f = 2$ are considered, as depicted in Figure 3. Here, Equation (53) is used for evaluating the dimensionless wave force. Figure 6 shows the dimensionless wave force $K_F$ varying against the dimensionless wavenumber $k_{1,0}h_1$ for different numbers of evanescent modes $N$. Convergences can also be observed to be significant. In addition, the results also agree well with those in the literature [21].

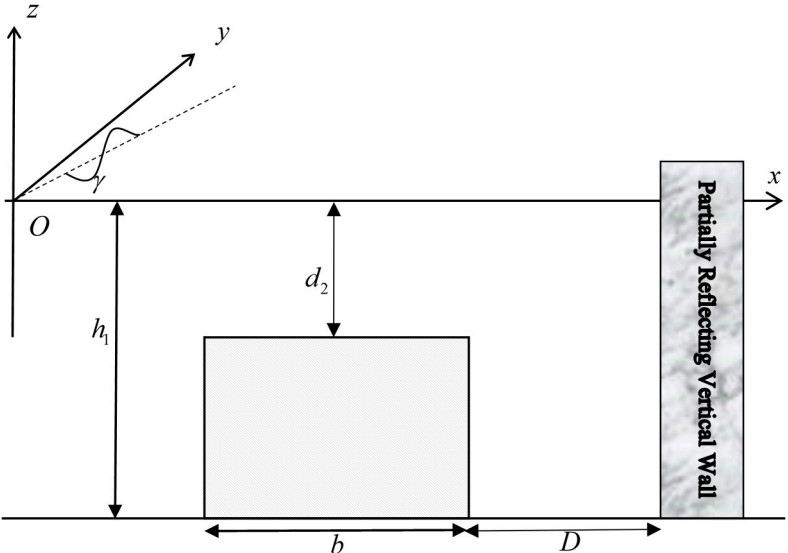

**Figure 3.** Problem definition of water wave scattering by a rectangular porous breakwater near a partially reflecting vertical wall over a uniform bottom.

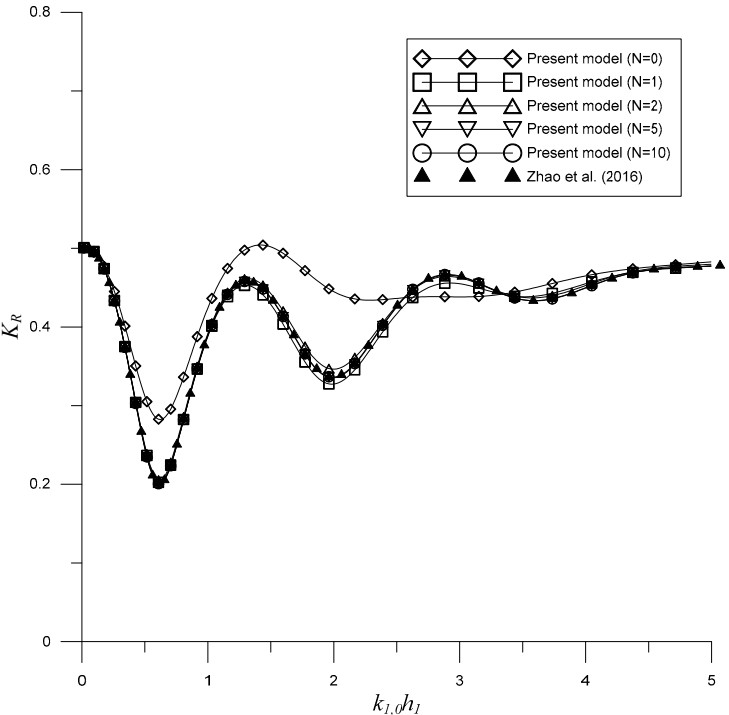

**Figure 4.** Reflection coefficient varying against dimensionless wavenumber for normally incident water wave scattering by a rectangular porous breakwater near a partially reflecting vertical wall.

**Table 1.** Convergences of $K_R$ with the number of evanescent modes $N$.

| $N$ | $k_{1,0}h_1 = \pi/10$ | $k_{1,0}h_1 = \pi/3$ | $k_{1,0}h_1 = \pi$ |
|---|---|---|---|
| 0 | 0.4167 | 0.4419 | 0.4387 |
| 1 | 0.3960 | 0.4070 | 0.4499 |
| 2 | 0.3962 | 0.4110 | 0.4555 |
| 3 | 0.3954 | 0.4085 | 0.4550 |
| 5 | 0.3953 | 0.4087 | 0.4560 |
| 10 | 0.3952 | 0.4088 | 0.4566 |
| 15 | 0.3952 | 0.4087 | 0.4566 |

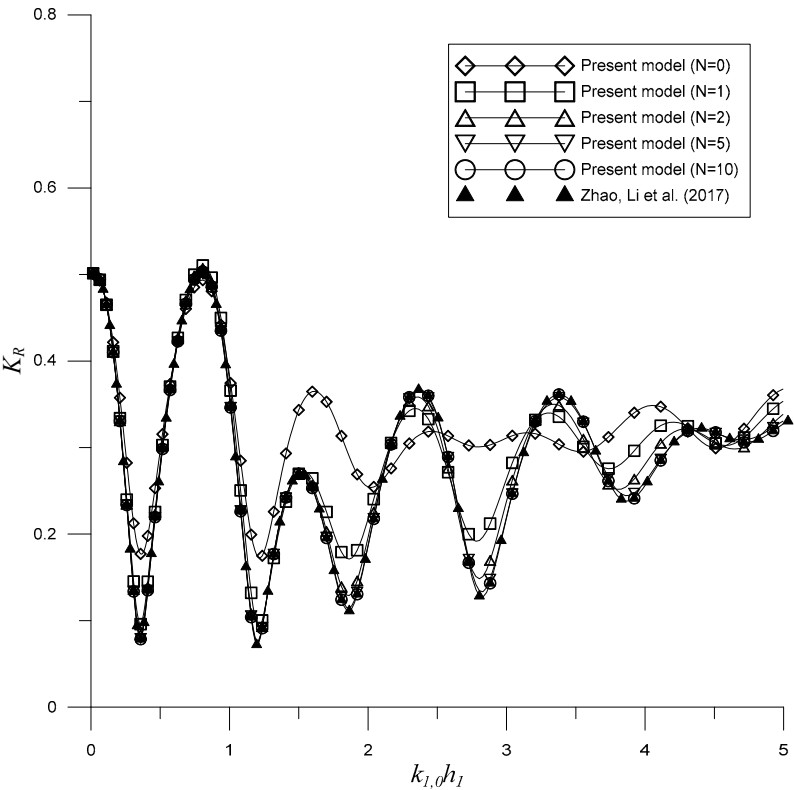

**Figure 5.** Reflection coefficient varying against dimensionless wavenumber for obliquely incident water wave scattering by a rectangular porous breakwater near a partially reflecting vertical wall.

Then the wave forces acting on the vertical wall with different partially reflecting factors $K_w$ are considered. The problem is also defined by Figure 3 with $\gamma = 0$, $D/h_1 = 3$, $b/h_1 = 1.2$, $d_2/h_1 = 0.5$, $\varepsilon = 0.4$, and $f = 2$. In Figure 7, the dimensionless wave forces $K_F$ are plotted against the dimensionless wavenumber $k_{1,0}h_1$ for different partially reflecting factors $K_w$ of the vertical wall. In the figure, good agreements with convergences between the present results and those of Zhao et al. [21] can be observed. Additionally, it can be observed that the resulted wave forces are larger for the vertical walls with the larger partially reflecting factors.

Overall, the results in this subsection indicates that the proposed EMM can evaluate the wave force on the partially reflecting vertical wall well.

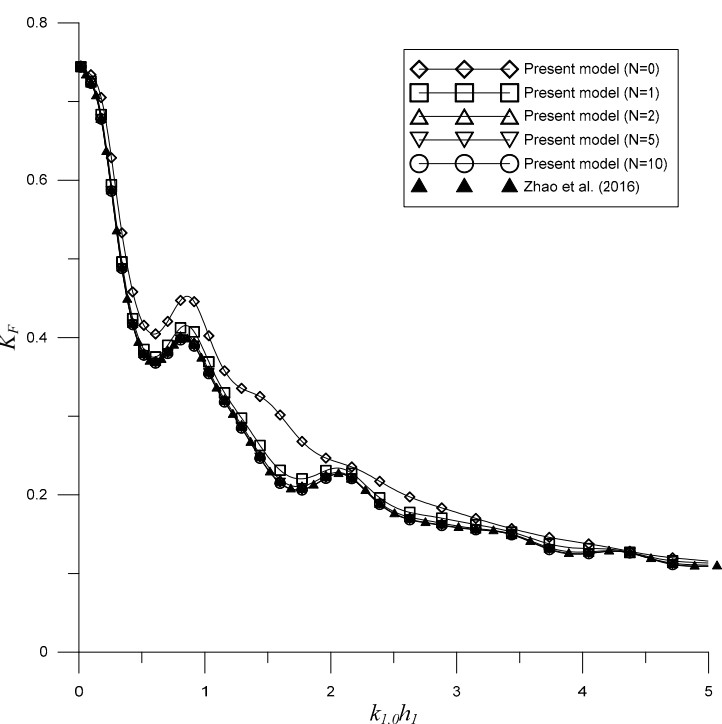

**Figure 6.** Dimensionless wave force varying against dimensionless wavenumber for normally incident water wave scattering by a rectangular porous breakwater near a partially reflecting vertical wall.

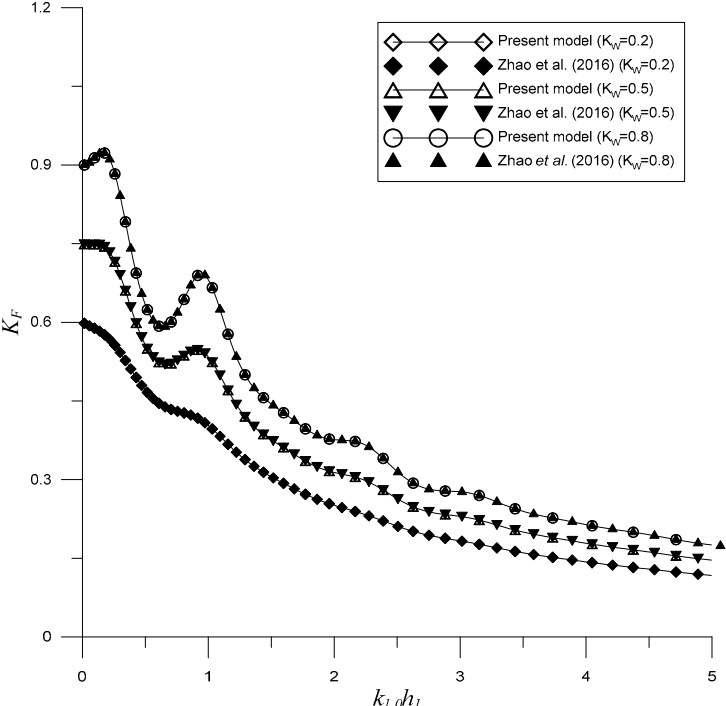

**Figure 7.** Dimensionless wave force varying against dimensionless wavenumber for normally incident water wave scattering by a rectangular porous breakwater near a partially reflecting vertical wall with different partially reflecting factors.

*3.3. Multiple Porous Structures near a Totally Reflecting Vertical Wall*

Then, the EMM is applied for solving water wave scattering by multiple rectangular porous breakwaters near a totally reflecting vertical wall ($K_w = 1$) as depicted in Figure 8.

In the figure, $L$ is the number of rectangular porous breakwaters and $h_1$ is the total depth. Furthermore, the porous parameters are set as $\varepsilon = 0.4$ and $f = 2$ in this subsection.

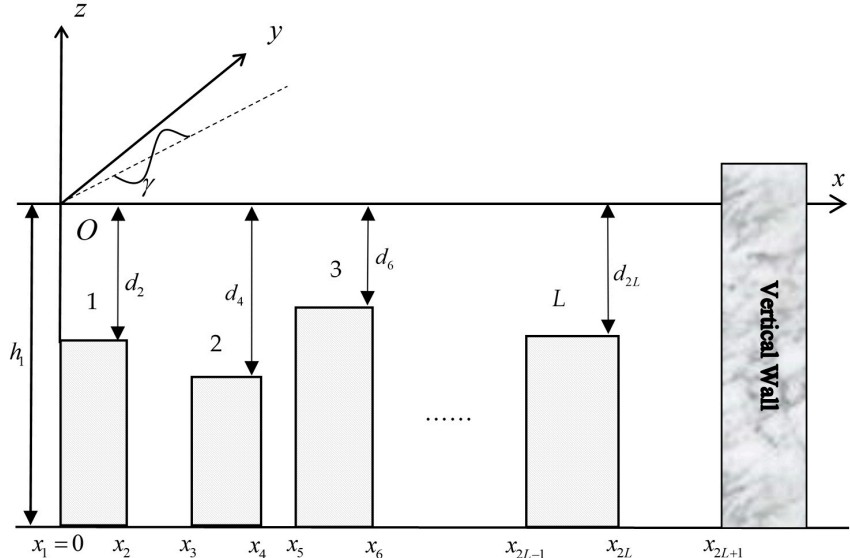

**Figure 8.** Problem definition of water wave scattering by multiple rectangular porous breakwaters near a vertical wall over a uniform bottom.

Following Zhao et al. [56], the monochromatic wave is considered to have an oblique incidence angle $\gamma = 30°$ and three rectangular porous breakwaters are under the dimensionless water depths $d_2/h_1 = 0.6$, $d_4/h_1 = 0.2$, and $d_6/h_1 = 0.4$. Additionally, the horizontal scales are $x_2/h_1 = 0.5$, $x_3/h_1 = 2.5$, $x_4/h_1 = 3.1$, $x_5/h_1 = 7.1$, $x_6/h_1 = 7.9$, and $x_7/h_1 = 10.9$. Figure 9 depicts the reflection coefficient $K_R$ varying against the dimensionless wavenumber $k_{1,0}h_1$ for different numbers of evanescent modes $N$. Convergences can also be observed to be significant. In addition, the results also agree well with those in the literature.

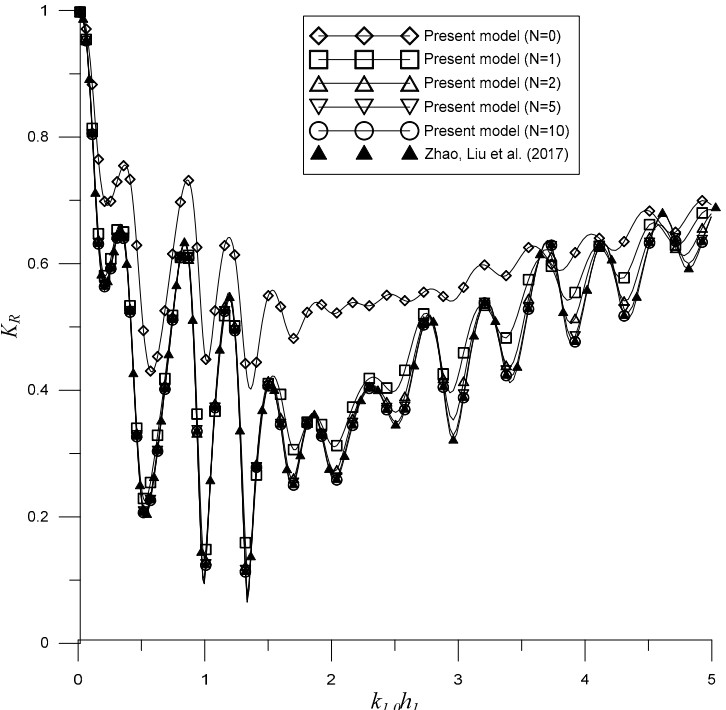

**Figure 9.** Reflection coefficient varying against dimensionless wavenumber for obliquely incident water wave scattering by three porous breakwaters near a totally reflecting vertical wall.

Then, combined effects of the Bragg scattering by six rectangular porous breakwaters and the totally reflecting vertical wall are examined. The six rectangular porous breakwaters have vertical scale $d_{2l}/h_1 = 0.5$ and horizontal scales $x_{2l-1}/h_1 = 2.2(l-1)$ & $x_{2l}/h_1 = 2.2(l-1) + 0.2$ for $l = 1, 2, \ldots, 6$. Furthermore, the totally reflecting vertical wall is located at $x_{13}/h_1 = 14.7$. In Figure 10, the reflection coefficient $K_R$ are plotted against the dimensionless wavenumber $k_{1,0}h_1$ for different numbers of evanescent modes $N$ with significant convergences. Also, the results also agree well with those in the literature [56]. More detailed studies of Bragg scattering by multiple variable porous breakwaters with a partially reflecting vertical wall will be conducted in the next section.

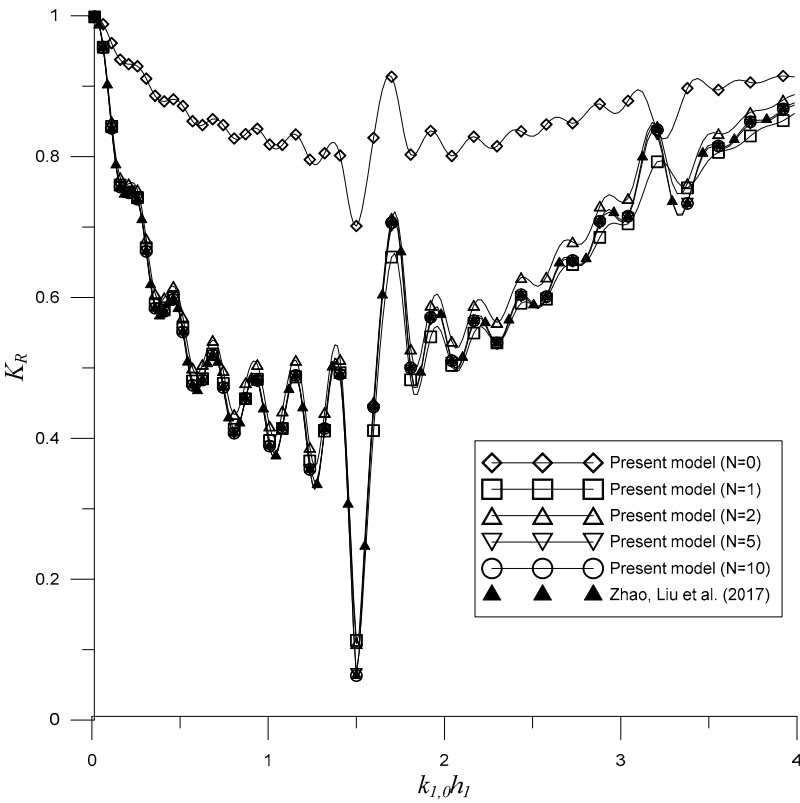

**Figure 10.** Reflection coefficient varying against dimensionless wavenumber for obliquely incident water wave scattering by six porous breakwaters near a totally reflecting vertical wall.

In summary, the results in this section demonstrates that the EMM can be applied for solving water wave scattering by multiple rectangular porous breakwaters near a vertical wall.

### 3.4. Trapezoidal Porous Breakwaters near a Porous Seawall

The main strength of the proposed EMM is the ability for solving problems with variable porous breakwaters. Behera and Khan [23] used the boundary element method (BEM) for solving water wave scattering by a singular breakwater or double trapezoidal porous breakwaters near porous seawalls as depicted in Figures 11 and 12, respectively. Here, the porous seawall is composed of a porous structure attached to a totally reflecting vertical wall. In this subsection, the EMM is applied for solving the prescribed problems and comparisons are conducted.

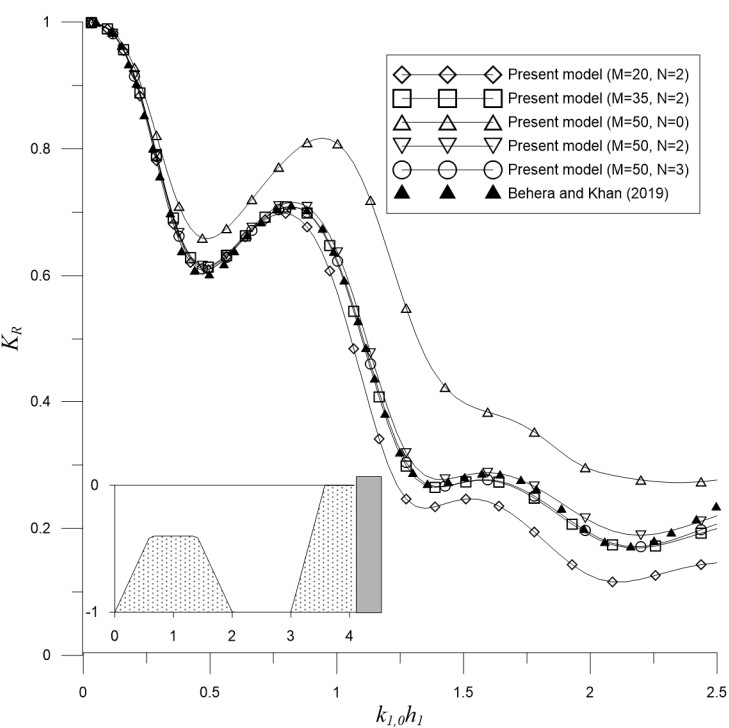

**Figure 11.** Reflection coefficient varying against dimensionless wavenumber for obliquely incident water wave scattering by a single trapezoidal porous breakwater near a porous seawall.

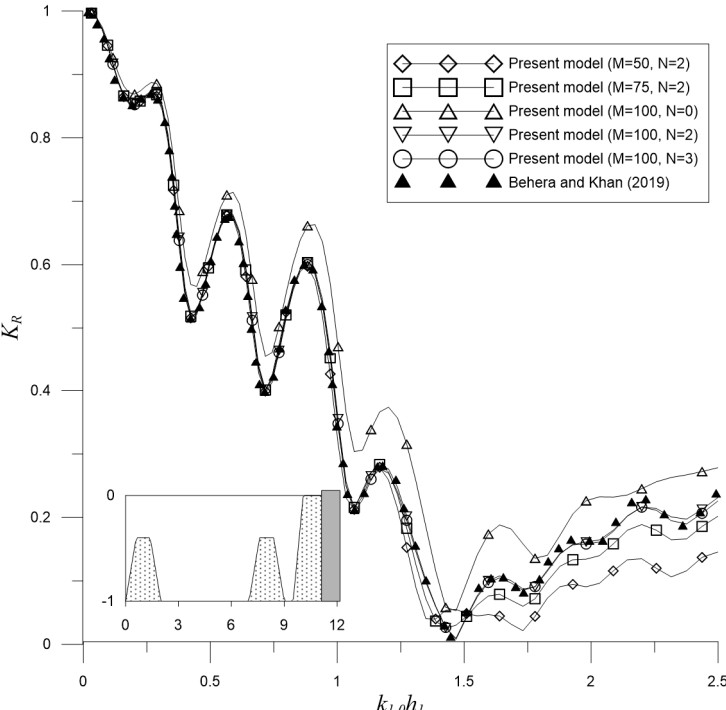

**Figure 12.** Reflection coefficient varying against dimensionless wavenumber for obliquely incident water wave scattering by two trapezoidal porous breakwaters near a porous seawall.

In the following of this subsection, the trapezoidal porous breakwaters are with the dimensionless bottom width and height being equal to 2 and 0.6, respectively. Both of the porous breakwaters and the porous seawall are of porosity $\varepsilon = 0.417$. If two breakwaters are considered, the dimensionless horizontal distance between the centers of the breakwaters

is 7. The side slopes are 1 and $\sqrt{3}$ for the trapezoidal breakwaters and the porous seawall, respectively. Additionally, the incidence angle is set as $\gamma = 10°$.

Firstly, the case of a single trapezoidal porous breakwater with $f = 1$ is considered. The dimensionless seawall top width and the separation distance between the toes of the breakwater and seawall are 0.5 and 1, respectively. In Figure 11, the reflection coefficient $K_R$ are plotted against the dimensionless wavenumber $k_{1,0}h_1$ for different numbers of evanescent modes $N$ and steps. In the figure, convergent results can be obtained with $N = 2$ and $M = 50$. Fewer evanescent modes are required for this case since the bottom is smoother. In addition, the results are also in good agreements with those in the literature [23].

Secondly, the case of two single trapezoidal porous breakwaters with $f = 0.5$ is considered. The dimensionless seawall top width and the separation distance between the toes of the breakwater and seawall are 1 and 0.5, respectively. Figure 12 gives the reflection coefficient $K_R$ varying against the dimensionless wavenumber $k_{1,0}h_1$ for different numbers of evanescent modes $N$ and steps. In the figure, $N = 2$ and $M = 100$ are required to have convergent results and good agreements with the BEM results [23]. When compared with the previous case, more steps are required since two trapezoidal porous breakwaters are considered.

Therefore, the results in this subsection indicate that the EMM can be applied to solve problems of water wave scattering by variable porous breakwaters near a porous seawall, which is composed of a porous structure attached to a totally reflecting vertical wall.

## 4. Discussion

After the EMM model is validated, it is applied for studying oblique Bragg scattering by four periodic half-cosine shaped breakwaters near a partially reflecting wall over uneven bottoms as depicted in Figure 13. The four periodic breakwaters are considered to be either porous, partially porous, or impermeable. Following Kirby and Anton [65] and Tsai et al. [66], the water depth and the amplitude of the half-cosine shaped breakwaters are set as $h_1 = 0.15$ m and $a = 0.05$ m, respectively. Furthermore, the wavelength of the periodic bottom and the separation distance between half-cosine breakwaters are defined by $2\pi/K = 0.8$ m and $b = 0.5$ m, respectively. The vertical wall with a partially reflecting factor $K_w$ is separated by $D$ away from the toe of the last breakwater.

Before studying the physical phenomena of the constructive and destructive Bragg scattering, convergences with respect to the numbers of steps and evanescent modes are studied. Here, the problem of Bragg scattering by four periodic half-cosine shaped impermeable breakwaters without vertical wall is first considered. In Figure 14, the convergence can be found for $N = 5$ and $M = 200$. In addition, the corresponding results are in good agreements with those of Kirby and Anton [65]. Additionally, the primary and secondary Bragg resonances can be observed to be significant at $2k_{1,0}/K \sim 1$ and $2k_{1,0}/K \sim 2$, respectively. Therefore, $N = 5$ and $M = 200$ will be used in the following of this section.

### 4.1. Constructive Bragg Scattering by the Partially Reflecting Wall

If a totally reflecting wall is located in the right end of periodic impermeable structures, the reflection coefficient of the problem will be approximately equal to one according to the conservation of energy. However, in practical coastal environments, the shoreline is occasionally protected by vertical breakwaters or seawalls which are usually considered to be partially reflecting.

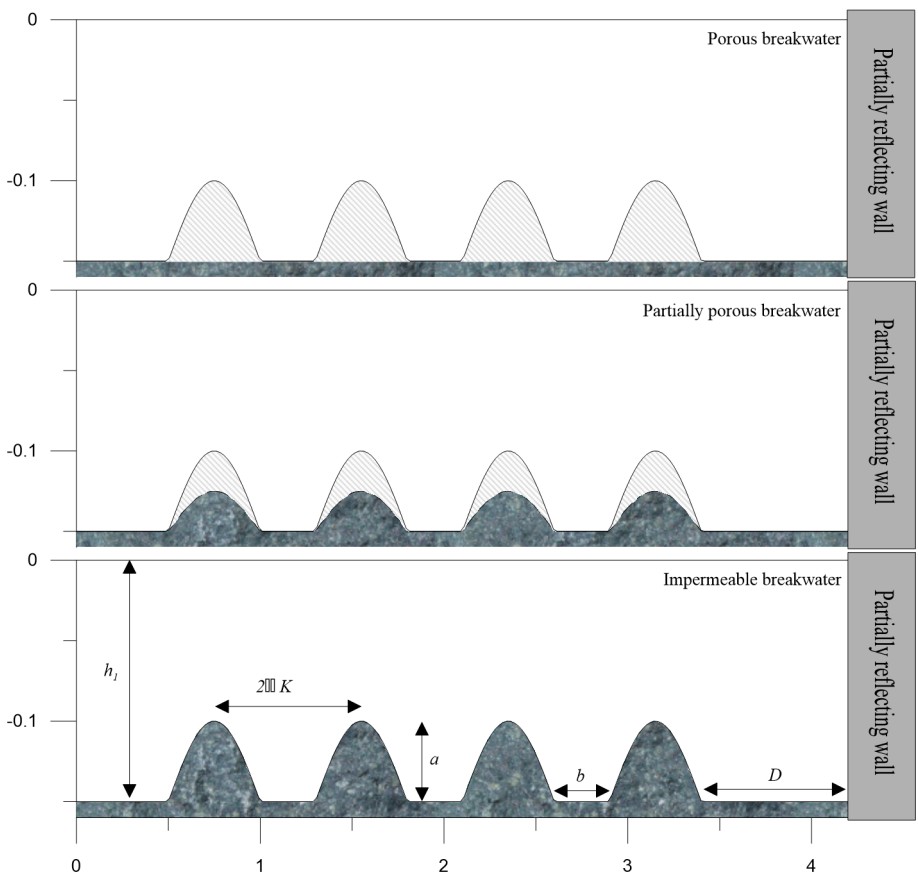

**Figure 13.** Problem definition of Bragg scattering by (**up**) porous; (**middle**) partially porous; (**down**) impermeable half-cosine breakwaters near a partially reflecting vertical wall over a uniform bottom.

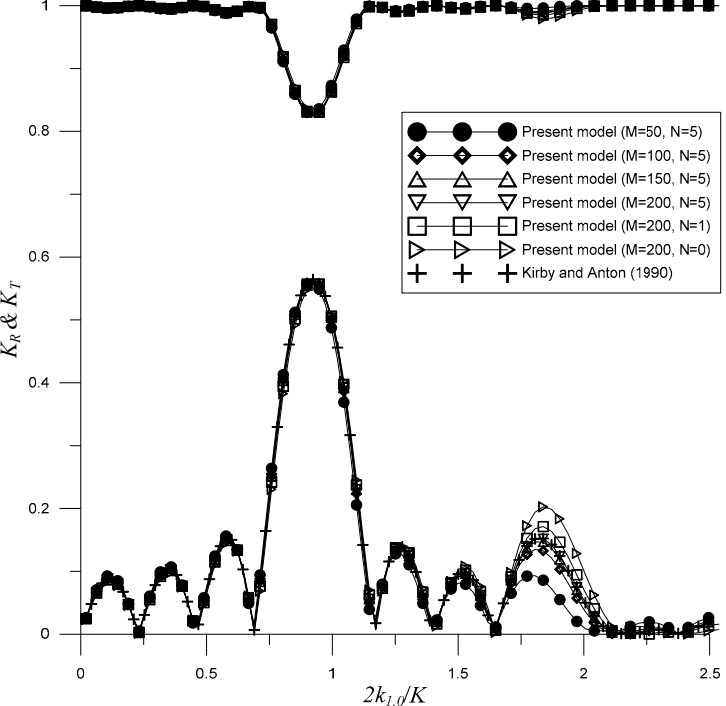

**Figure 14.** Reflection and transmission coefficients varying against $2k_{1,0}/K$ for Bragg scattering by four periodic half-cosine shaped impermeable breakwaters without vertical wall.

According to our preliminary studies, the separation distance between the last break-water and the vertical wall (Figure 13) is considered to be

$$D = \frac{(2p-1)}{2}\frac{2\pi}{K} = 0.4(2p-1)m \tag{54}$$

for $p = 1, 2, \ldots$. For $D = 0.4$ m (or $p = 1$), Figure 15 depicts the reflection coefficient and dimensionless wave force obtained by the EMM model. In the figure, it can be observed that both the primary and secondary Bragg resonances are constructive. Additionally, the wave forces are migrated significantly at the primary Bragg resonance for all three cases with different reflection coefficients of the vertical wall.

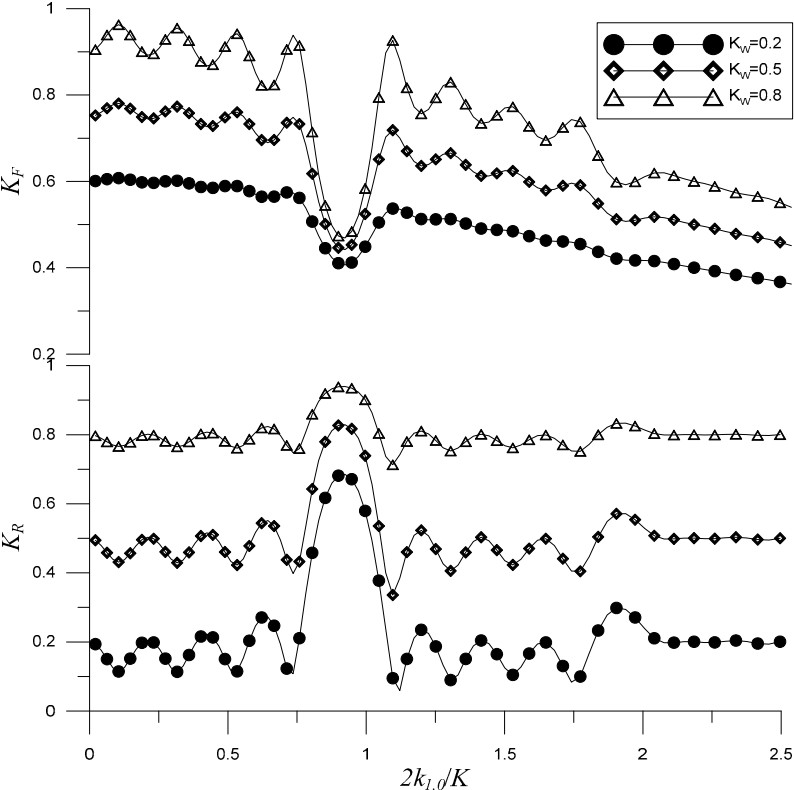

**Figure 15.** Reflection coefficient and dimensionless wave force varying against $2k_{1,0}/K$ with $D = 0.4$ m for normal-wave Bragg scattering by four periodic half-cosine shaped impermeable breakwaters with partially reflecting vertical wall.

Then, we also consider the configuration with $D = 1.2$ m (or $p = 2$). The reflection coefficient and dimensionless wave force can also be solved by the EMM model as described in Figure 16. In the figure, it is obvious that the configuration results in constructive primary Bragg resonances with destructive secondary ones. Detailed reason will be investigated in our future studies.

These results suggest that multiple breakwaters should be periodically located with the wavelength equal to half of the significant wavelength of the coastal wave environment. In addition, the partially vertical wall should be located by a quarter of the significant wavelength away from the last breakwater. In this situation, the wave forces on the wall can be significantly reduced.

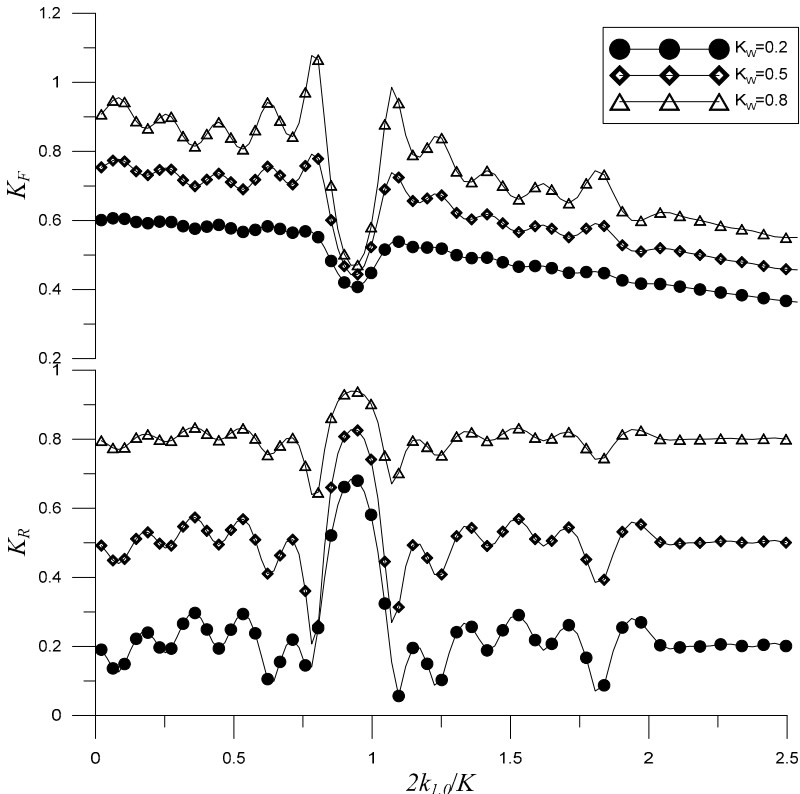

**Figure 16.** Reflection coefficient and dimensionless wave force varying against $2k_{1,0}/K$ with $D = 1.2$ m for normal-wave Bragg scattering by four periodic half-cosine shaped impermeable breakwaters near a partially reflecting vertical wall.

### 4.2. Destructive Bragg Scattering by the Partially Reflecting Wall

Then, the destructive Bragg scattering by the partially reflecting wall are also considered. Similarly, the preliminary studies suggest that

$$D = q\frac{2\pi}{K} = (0.8q)m \tag{55}$$

for $q = 1, 2, \ldots$.

For $D = 0.8$ m (or $q = 1$), Figure 17 shows the reflection coefficient and dimensionless wave force obtained by the EMM model. In the figure, it is significant to observe that both the primary and secondary Bragg resonances are destructive. Therefore, the results indicate that the separation distance should not be equal to the wavelength of the periodical bottom so that the extreme wave forces on the vertical wall can be avoided.

### 4.3. Oblique Incidence

In practical coastal environments, the significant wave angle, $\gamma$, is sometimes oblique to the normal direction of the shoreline or the protecting vertical wall. Following Mei [67] and Dalrymple et al. [68], the wavenumber of the bottom structures, $K$, should be set as $2k_{1,0}\cos\gamma/K \sim 1$ so that the Bragg scattering can be used to protect the coast. In this study, we also found that Equation (54) should be adopted to locate the partially reflecting vertical wall so that the constructive primary Bragg scattering occurs.

Considering $\gamma = 30°$ and $D = 0.4$ m, Figure 18 shows the reflection coefficient and dimensionless wave force obtained by the EMM model. In the figure, it is significant to observe that both the primary and secondary Bragg resonances are constructive at $2k_{1,0}\cos\gamma/K \sim 1$ and $2k_{1,0}\cos\gamma/K \sim 2$, respectively. This confirms the description addressed in the previous paragraph.

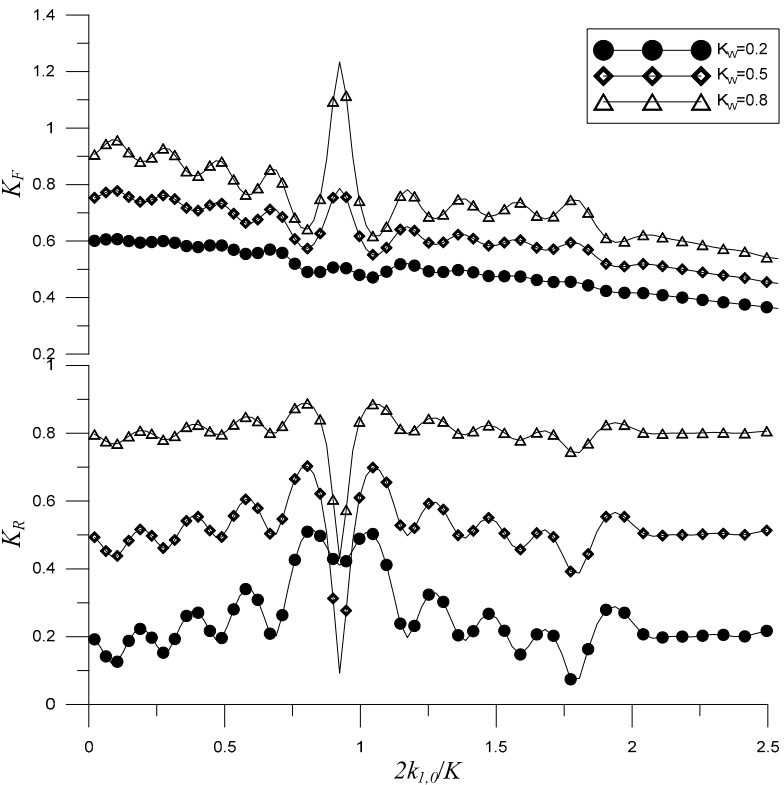

**Figure 17.** Reflection coefficient and dimensionless wave force varying against $2k_{1,0}/K$ with $D = 0.8$ m for normal-wave Bragg scattering by four periodic half-cosine shaped impermeable breakwaters near a partially reflecting vertical wall.

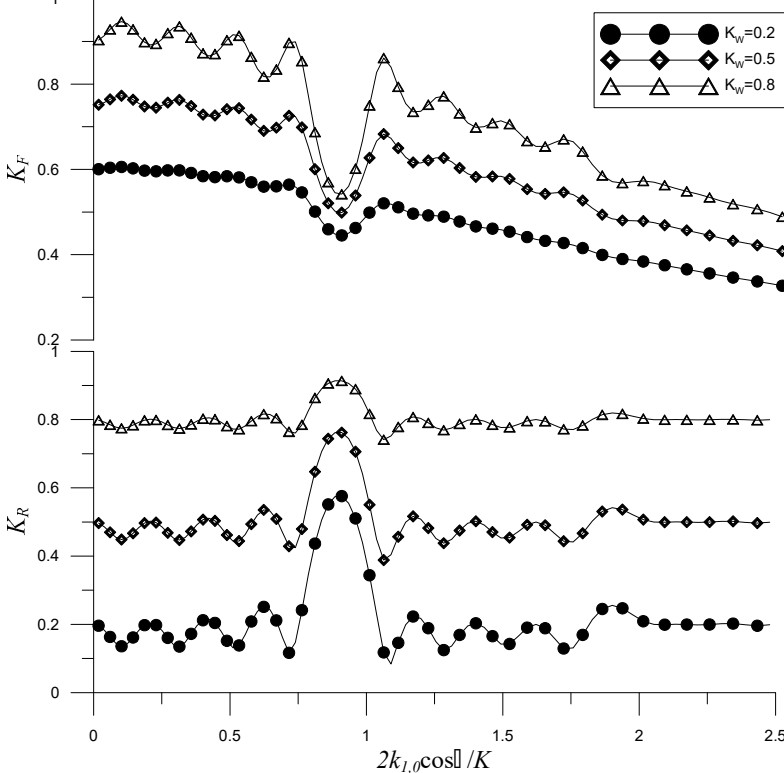

**Figure 18.** Reflection coefficient and dimensionless wave force varying against $2k_{1,0}cos\gamma/K$ with $D = 0.8$ m for oblique-wave Bragg scattering by four periodic half-cosine shaped impermeable breakwaters near a vertical wall with different partially reflecting factors.

Then, the wave directionality is studied by considering different incidence angles with the partially reflecting factor $K_w = 0.2$. Figure 19 describes the reflection coefficient and dimensionless wave force solved by the EMM model. This results further ensure the constructive Bragg scattering for oblique waves. Additionally, it can be observed that the wave forces are slightly smaller and the Bragg reflections are stronger for the cases with smaller incidence angles.

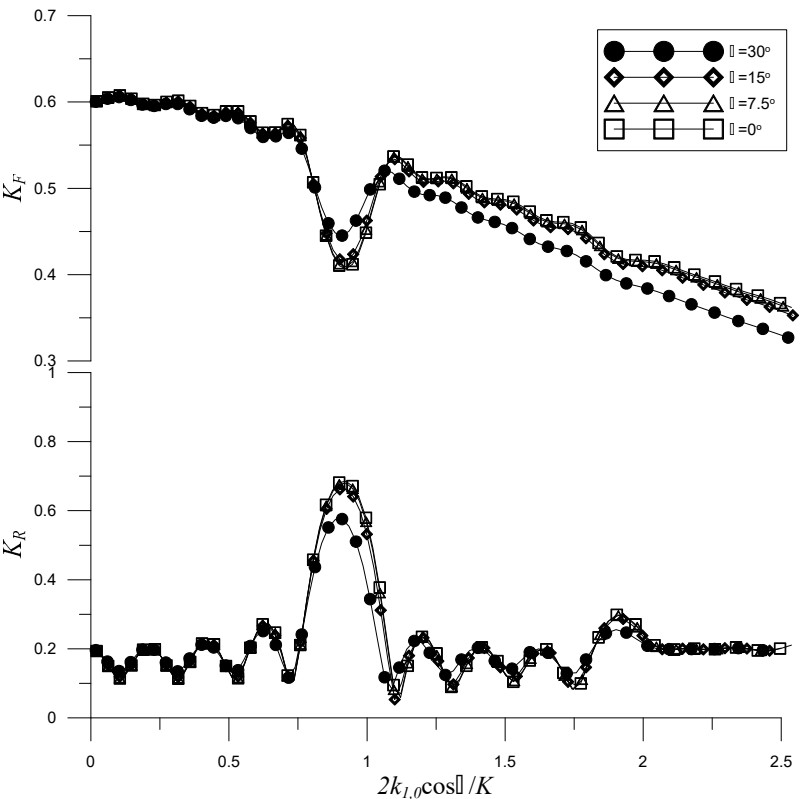

**Figure 19.** Reflection coefficient and dimensionless wave force varying against $2k_{1,0}cos\gamma/K$ with D = 0.8 m for oblique-wave Bragg scattering by four periodic half-cosine shaped impermeable breakwaters near a partially reflecting vertical wall with different incidence angles.

### 4.4. Periodic Porous Breakwaters

Then, the case of Figure 15 is re-examined by replacing the four impermeable breakwaters by either porous or partially porous breakwaters with $\varepsilon = 0.4$ and $f = 1$ as depicted in Figure 13 Additionally, $K_w = 0.5$ is set for all of the three cases.

Figure 20 shows the reflection coefficients and dimensionless wave forces obtained by the EMM model. In the figure, the constructive Bragg resonances can also be observed. Additionally, the use of porous breakwaters can further reduce the reflection coefficients and the wave forces. At the primary Bragg resonances, the wave forces of the three cases are of similarly magnitudes while the reflections are further reduced for the cases with porous breakwaters.

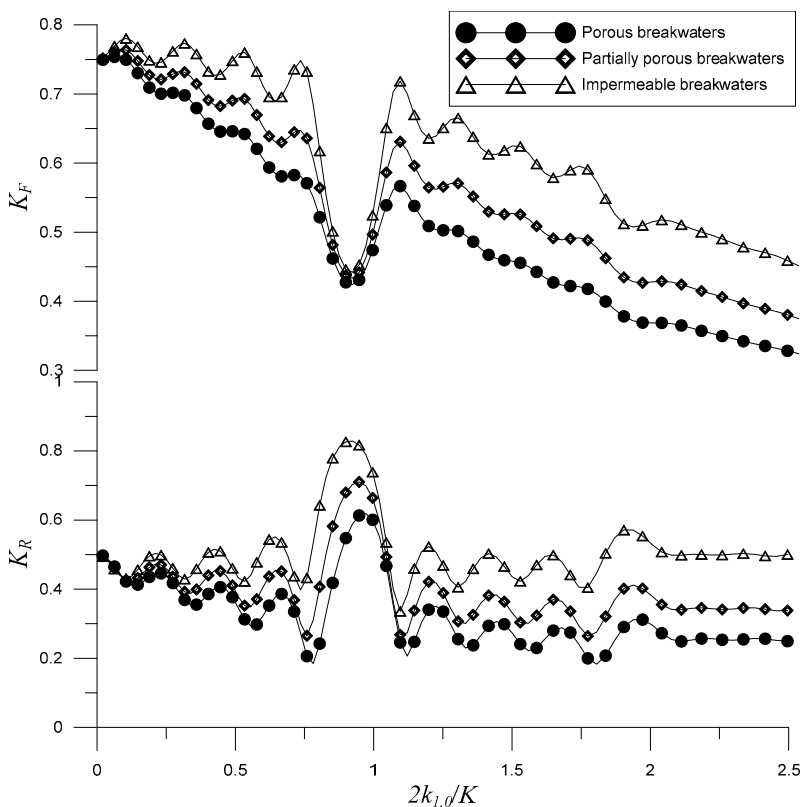

**Figure 20.** Reflection coefficient and dimensionless wave force varying against $2k_{1,0}/K$ with $D = 0.4$ m for normal-wave Bragg scattering by four periodic half-cosine shaped porous or impermeable breakwaters near a partially reflecting vertical wall.

## 5. Conclusions

In this article, the constructive and destructive Bragg scattering are studied for oblique water waves interactions with multiple variable porous/impermeable breakwaters near a partially reflecting wall over uneven bottoms by using the eigenfunction matching method (EMM). In the solution procedure, the variable breakwaters and bottom profiles are sliced into shelves separated steps and the solutions on the shelves are composed of eigenfunctions with unknown coefficients representing the wave amplitudes. Using the conservations of mass and momentum as well as conditions for the partially reflecting sidewall, a system of linear equations is resulted that can be solved by a sparse-matrix solver. Several cases are solved by the EMM to validate the proposed formulation. Then the constructive and destructive Bragg scattering for oblique water waves interactions with multiple variable porous/impermeable breakwaters near a partially reflecting wall over uneven bottoms are discussed. Numerical results indicated that the partially reflecting wall should be located by half of the wavelength of periodic breakwaters to migrate the wave forces on the vertical wall for both normal and oblique attacks of waves.

The proposed EMM is a depth-integrated model by assuming irrational, harmonic, and linear waves. The effects of wave breaking and dissipation can be included in the EMM formulation by the energy-dissipation factor of the MSE [69–71]. In addition, the proposed model is computationally efficient and can be served as preliminary calculations followed by modern three-dimensional numerical models. These are currently under investigation.

**Author Contributions:** Conceptualization, C.-C.T.; methodology, J.-Y.C. and C.-C.T.; software, C.-C.T.; writing—original draft, J.-Y.C.; visualization, J.-Y.C. All authors have read and agreed to the published version of the manuscript.

**Funding:** This research was funded by the Ministry of Science and Technology of Taiwan under the Grant No. MOST 109-2221-E-992-046-MY3.

**Institutional Review Board Statement:** Not applicable.

**Informed Consent Statement:** Not applicable.

**Acknowledgments:** The Ministry of Science and Technology of Taiwan is gratefully acknowledged for providing financial support to carry out the present work.

**Conflicts of Interest:** The authors declare no conflict of interest exist.

## Nomenclature

| | |
|---|---|
| $\alpha_{1,m,n}$ | coefficient of vertical eigenfunction |
| $\alpha_{2,m,n}$ | coefficient of vertical eigenfunction |
| $\beta_{1,m,n}$ | coefficient of vertical eigenfunction |
| $\beta_{2,m,n}$ | coefficient of vertical eigenfunction |
| $\gamma$ | incidence angle |
| $\varepsilon$ | porosity of porous media |
| $\lambda$ | wavelength of incident wave |
| $\sigma$ | angular frequency of incident wave |
| $\rho$ | density of water |
| $\phi_m$ | velocity potential on the $m-th$ shelf |
| $\eta_m$ | surface elevation on the $m-th$ shelf |
| $\delta$ | parameter for porous layer |
| $\theta_R$ | phase angle |
| $\zeta_{m,n}(z)$ | vertical eigenfunction |
| $\xi_{m,n}^{(1)}(x)$ | the first horizontal eigenfunction |
| $\xi_{m,n}^{(2)}(x)$ | the second horizontal eigenfunction |
| $\zeta_{m,l}^{larger}$ | vertical eigenfunction for the larger total depth |
| $\zeta_{m,l}^{smaller}$ | vertical eigenfunction for the smaller total depth |
| $\nabla$ | three-dimensional gradient operator |
| $\nabla^2$ | three-dimensional Laplace operator |
| $\Delta$ | operator for porous pressure |
| $\bar{a}$ | amplitude of incident wave |
| $a$ | amplitude of the half-cosine shaped breakwater in Section 4 |
| $b$ | width of the rectangular porous breakwater in Sections 3.1 and 3.2 or separation distance between half-cosine breakwaters in Section 4 |
| $d_m$ | water depth on the $m-th$ shelf |
| $f$ | friction coefficient of porous media |
| $g$ | acceleration of gravity |
| $h_m$ | total depth on the $m-th$ shelf |
| $\mathbf{i}$ | unit of complex numbers |
| $k_y$ | transverse wavenumber |
| $\hat{k}_{m,n}$ | lateral wavenumber of the $n-th$ evanescent mode on the $m-th$ shelf |
| $k_{m,n}$ | absolute wavenumber of the $n-th$ evanescent mode on the $m-th$ shelf |
| $\hat{k}_{m,0}$ | lateral wavenumber of the propagating mode on the $m-th$ shelf |
| $k_{m,0}$ | absolute wavenumber of the propagating mode on the $m-th$ shelf |
| $n$ | index of modes |
| $m$ | index for shelves and steps |
| $p_m$ | pressure on the $m-th$ shelf |
| $p$ | index for constructive Bragg scattering |
| $q$ | index for destructive Bragg scattering |
| $t$ | time |
| $s$ | inertial coefficient of porous media |
| $\mathbf{u}_m$ | fluid velocity or discharge velocity on the $m-th$ shelf |
| $x_m$ | x coordinate of the $m-th$ step |
| $\bar{x}_m$ | reference location of the |
| $(x, y, z)$ | three-dimensional Cartesian coordinates |
| $A_{m,n}$ | EMM unknown coefficients |

| $B_{m,n}$ | EMM unknown coefficients |
| --- | --- |
| $D$ | Separation distance between the last porous breakwater and the vertical wall in Sections 3.1, 3.2 and 4. |
| $G_1$ or $G_2$ | variable for depth eigenfunction |
| $K_w$ | partially reflecting factor of the vertical wall |
| $K_R$ | reflection coefficient |
| $K_F$ | dimensionless horizontal wave force on the vertical wall |
| $M$ | number of shelves plus one |
| $N$ | number of evanescent modes |
| $T$ | wave period of incident wave |
| $2\pi/K$ | wavelength of the periodic bottom in Section 4 |

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
