# Peer review of "Wave Forces on a Partially Reflecting Wall by Oblique Bragg Scattering with Porous Breakwaters over Uneven Bottoms"

_jmse, doi:10.3390/jmse10030409_

Round 1
Reviewer 1 Report
The paper is well written and organized. It can be accepted for publication in JMSE with very minor revision as follows:
(1) Please add some more important observations in the Conclusion.
(2) The manuscript is missing some recent references to the similar physical models:
Boundary element method for wave trapping by a multi-layered trapezoidal breakwater near a sloping rigid wall. Meccanica, 56(2), 317 - 334 https://doi.org/10.1007/s11012-020-01286-z
Analysis of wave action through multiple submerged porous structures. Journal of Offshore Mechanics and Arctic Engineering, ASME , 142(1), 011101 https://doi.org/10.1115/1.4044360
Wave attenuation properties of rubble mound breakwater in tandem with a floating dock against oblique regular waves https://doi.org/10.1080/17455030.2021.1967512
Author Response
As attached.

Reviewer 2 Report
The comments are attached.

Author Response
As attached

Reviewer 3 Report
Please see the attached word file.

Author Response
As attached

Round 2
Reviewer 2 Report
The authors have made significant revision. I accept the revised paper for its publication in Journal of Marine Science and Engineering, MDPI.
Author Response
Many thanks for your review report.
Reviewer 3 Report
The authors have addressed the majority of the comments of the reviewer by (i) adding clarifications, (ii) expanding the introduction section and providing more information in the manuscript, (iii) generating a ‘Nomenclature” section, and (iv) including a new figure about the wave directionality effect.
Overall, the aforementioned changes increased the size of the manuscript and improved its quality, increasing its value for the reader. This is a testament of the authors’ sincere effort to address diligently the comments of the reviewers. So, thank you for your effort.
There are only a few minor comments, as shown below. Therefore, the manuscript is suggested to be accepted for publication with minor revisions.
Minor comments:
Line 55-56: Replace “…the prescribed property depends on the width of the wall” with “the hydrodynamic forces depend on the ratio of the wavelength-to-width of the coastal structure”
Line 63: Replace “reducing wave force” with “reducing the wave force”
Line 101: Replace “of MSE solutions” with “of the MSE solutions”
Line 104: Replace “investigations” with “investigation”
Line 226-227: Replace “by the methods for accessing the physical and computational data [60-62]” with “by physical or other computational methods [60-62]”
Line 308: Replace “researches” with “research”
Line 362: Replace “are attacked by larger wave forces” with “are attracting larger wave forces”. (please note that generally the phrase “are attacked” can refer to waves, but not to wave forces).
Line 391: Replace “agree well those in the literature” with “agree well with those in the literature”
Author Response
As attached.
